# Riboswitch Mechanisms for Regulation of P1 Helix Stability

**DOI:** 10.3390/ijms251910682

**Published:** 2024-10-04

**Authors:** Jason R. Stagno, Yun-Xing Wang

**Affiliations:** Protein-Nucleic Acid Interaction Section, Center for Structural Biology, Center for Cancer Research, National Cancer Institute, Frederick, MD 21701, USA; wangyunx@mail.nih.gov

**Keywords:** riboswitch, RNA structure, regulatory RNA, aptamer, P1 helix

## Abstract

Riboswitches are highly structured RNA regulators of gene expression. Although found in all three domains of life, they are particularly abundant and widespread in bacteria, including many human pathogens, thus making them an attractive target for antimicrobial development. Moreover, the functional versatility of riboswitches to recognize a myriad of ligands, including ions, amino acids, and diverse small-molecule metabolites, has enabled the generation of synthetic aptamers that have been used as molecular probes, sensors, and regulatory RNA devices. Generally speaking, a riboswitch consists of a ligand-sensing aptamer domain and an expression platform, whose genetic control is achieved through the formation of mutually exclusive secondary structures in a ligand-dependent manner. For most riboswitches, this involves formation of the aptamer’s P1 helix and the regulation of its stability, whose competing structure turns gene expression ON/OFF at the level of transcription or translation. Structural knowledge of the conformational changes involving the P1 regulatory helix, therefore, is essential in understanding the structural basis for ligand-induced conformational switching. This review provides a summary of riboswitch cases for which ligand-free and ligand-bound structures have been determined. Comparative analyses of these structures illustrate the uniqueness of these riboswitches, not only in ligand sensing but also in the various structural mechanisms used to achieve the same end of regulating switch helix stability. In all cases, the ligand stabilizes the P1 helix primarily through coaxial stacking interactions that promote helical continuity.

## 1. Introduction

Riboswitches, discovered more than 20 years ago [1,2], are segments of RNA found in the non-coding regions of many mRNAs, whose complex three-dimensional (3D) folds function as inducible molecular switches for the *cis*-regulation of gene expression. The switch mechanism involves allosteric changes that occur in response to the binding of a cognate ligand to the ligand-sensing domain (aptamer). Riboswitches are named and classified according to the ligand that they selectively sense, which includes a variety of metabolites, elemental ions, and cofactors. As of 2022, a total of 56 distinct classes of naturally occurring riboswitches had been experimentally validated [3,4], with several others proposed or under current investigation. Depending on the riboswitch, genetic regulation is achieved at the level of transcription termination or translation initiation, or, at least in one identified case, RNA splicing [5].

Of the nearly five dozen classes of riboswitches discovered over the last two decades, only one class (TPP riboswitches) has been identified in Eukarya, and none have been found in animals. For this reason, many riboswitches, particularly those found in pathogenic bacteria, have been proposed as strategic targets for the development of novel antibiotics, but such an approach has yet to be proven viable. However, due to their structural diversity and versatility, riboswitch aptamers have been used in a variety of molecular applications and synthetic engineering platforms. Using evolutionary in vitro selection methods, namely SELEX [6], novel synthetic aptamers can be generated to target a small molecule of choice. In addition, both naturally occurring and synthetic fluorescent aptamers have been employed as molecular markers and biosensors [7,8]. Other biotechnology applications include the conjugation of aptamers to therapeutic agents or drug delivery systems [9] or to ribozymes to form inducible self-cleaving RNA devices (aptazymes) [10].

In most cases, the structure of a riboswitch comprises two domains: the aptamer and the expression platform. Ligand binding to the aptamer domain triggers a conformational switch, ultimately leading to secondary structural changes in the expression platform that control the fate of gene expression. For many riboswitches, the underlying structural mechanism for this regulatory behavior involves a short stretch of nucleotides (switching sequence) that can form two mutually exclusive secondary structures, whose formation or deformation is ligand-dependent. One of these regulatory structural elements is quite often the aptamer’s P1 helix, whereby ligand binding/unbinding regulates the stability of the duplex. Although regulatory switch helices are involved quite broadly across the gamut of riboswitches, this review focuses on P1 helix-regulated riboswitches, as defined in a previous review [11], which are some of the most widely studied and structurally characterized. Of note, many other riboswitches not discussed here may exhibit P1 helix-regulating behavior but have been excluded on the basis of architecture, such as those consisting of a single helical domain formed by strands that are adjacent in sequence (e.g., pre-quenosine-responsive riboswitches). Generally speaking, P1 helix-regulated riboswitches are those meeting the following characteristics: (1) the aptamer comprises two or more stem loops and a regulatory helix (P1), the latter being formed through the long-range base pairing of non-adjacent strands (typically the termini); (2) the helical domains of the aptamer intersect at an *n* > 2 junction, the structure and stability of which involves ligand binding; (3) in a ligand-controlled manner, the 3′ strand of P1 forms a mutually exclusive structure in the expression domain that regulates gene expression.

To understand the structural basis for conformational switching and the mechanism of the regulation of P1 helix stability, structural information about riboswitch aptamers in both apo and holo forms is required. However, due to their often inherent flexibility and structural instability in the absence of a ligand, there is great difficulty in the structural determination of aptamers in ligand-free (apo) states. X-ray crystallography has been the dominant method for the determination of the structures of aptamers. However, most crystallization conditions contain high concentrations of precipitants and are often of high ionic strength. Even in the absence of a ligand, such conditions, coupled with crystal packing forces, can induce conformational changes and ultimately trap the aptamer in a state that resembles (or is virtually identical to) the ligand-bound (holo) conformation. This challenge becomes increasingly problematic for conditions containing Mg^2+^, which can have dramatic effects on RNA compaction, even at modest concentrations (e.g., 10 mM) [12].

Despite numerous riboswitches utilizing the P1 switch helix for the regulation of gene expression, the structural mechanism by which this is accomplished surprisingly varies. This review focuses on the limited number of P1 helix-regulated riboswitches for which apo and holo structures have been determined, which include both transcriptional and translational riboswitches from distinct classes. The comparison of these structures and their underlying switching mechanisms reveals the versatility of riboswitches to utilize not only different ligands but also different means to achieve the same end—regulating gene expression through ligand-controlled P1 helix stability.

## 2. Riboswitches with Structurally Elucidated P1-Stabilizing Mechanisms

### 2.1. Adenine Riboswitch

Perhaps considered the poster child of riboswitches, the purine class has long served as a model system for understanding the structural and biophysical properties of small-molecule-sensing aptamers, as well as the widely distributed three-way-junction (3WJ) architecture found among various types of RNAs [13,14]. Given its small size, the purine riboswitch exhibits a minimal structure containing all the representative features of the Y-shaped RNA fold: three intersecting helical domains partially stabilized by a distal tertiary interaction. The first structure of a purine riboswitch, the guanine riboswitch in complex with hypoxanthine, was determined in 2004. It is one of the earliest and highest-resolution riboswitch structures ever reported [15]. One month later, the structures of a guanine riboswitch and an adenine riboswitch, complexed with guanine and adenine, respectively, were also reported [16]. The purine riboswitches share a highly conserved sequence and three-dimensional architecture, whose nucleobase specificity is achieved primarily through a single ligand-sensing nucleotide: U74 for adenine and C74 for guanine/hypoxanthine/xanthine. The binding pocket is formed by the 3WJ, where the ligand is held in place in a base triple with junction residues, sandwiched between two adjacent base triples, thereby facilitating coaxial stacking between helices P1 and P3 (Figure 1).

In 2017, nearly 13 years later, ligand-free structures of a translation-regulating adenine riboswitch were reported that provided the structural basis for the regulation of P1 stability [17,18]. In contrast to the rigid and compact fold of the purine-bound aptamer, its conformation is quite flexible in the absence of a ligand. The crystal fortuitously contained two molecules in the asymmetric unit with unique unliganded conformations (apo1, apo2) that were significantly different from the ligand-bound conformation and from one another, most notably in the 3WJ [16,19]. The interconversion of apo1/apo2 (RMSD: 2.5 Å) enables an additional layer of thermodynamic control. In apo1, A23 of J1/2 (hinge region) mimics the ligand by intercalating between U48 and U49 (Figure 1A) and thus precludes ligand binding [18]. In apo2, the ligand position is occupied by U48, which forms weak interactions with U74 and a mobile residue (A21) in the flexible hinge region (Figure 1B). Thus, apo2 represents the binding competent conformation, in which the ligand can easily displace U48 to form an intermediate-bound conformer, followed by conversion to the holo conformation through an induced-fit mechanism.

**Figure 1 ijms-25-10682-f001:**
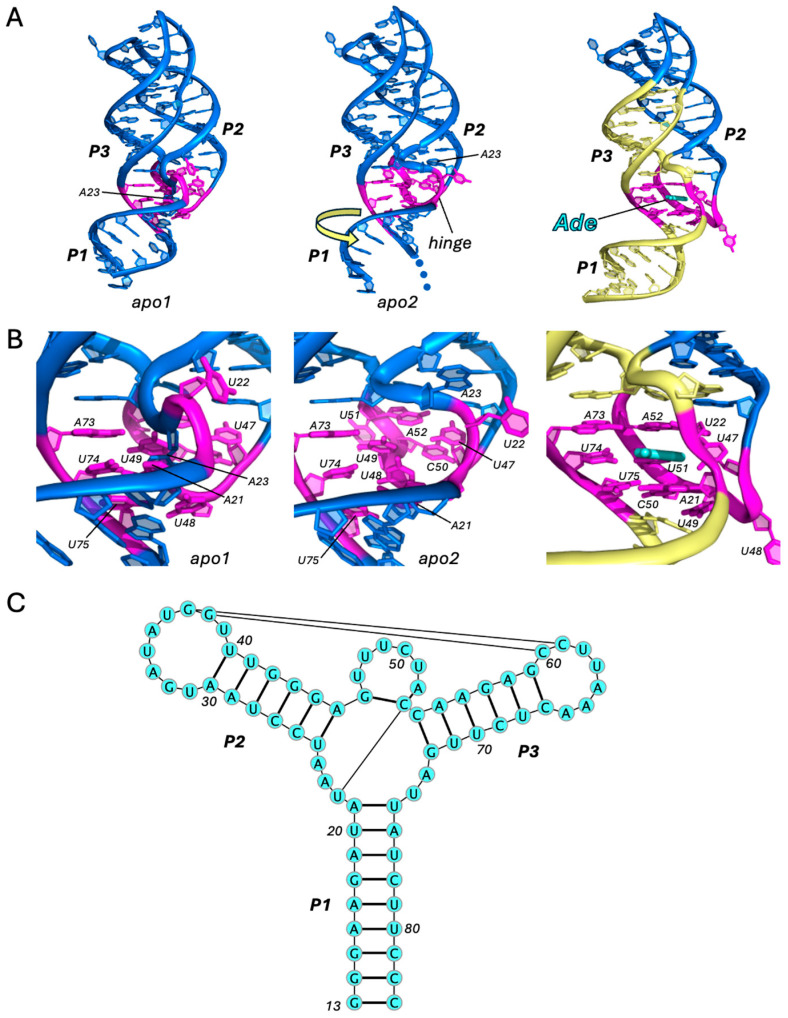
Adenine riboswitch. Comparison of the apo (PDB ID: 5E54) [18] and holo (PDB ID: 4TZX) [19] structures of the adenine riboswitch aptamer (**A**) and its ligand-binding site (**B**). All-atom RMSD between apo2 and holo structures is 3.6 Å. Adenine ligand is shown in cyan. Key residues are colored in magenta. Regions of coaxial stacking that stabilize P1 upon ligand binding are colored in yellow. Blue dots represent missing residues that are disordered in the crystal structure. Motion of conformational change from apo2 to holo states is indicated by the yellow arrow. See also Movie 1. (**C**) Secondary structure (PDB ID: 4TZX), generated using VARNA (v3-93) [20].

The absence of ligand-stabilizing interactions in the 3WJ disrupts the stack of base triples that facilitate the coaxial stacking of P1 and P3 (Figure 1B). The resulting flexibility of the junction induces the rotation and translation of P1 relative to P3 and the partial melting of the P1 duplex (Figure 1A). Thus, the mechanism for the regulation of P1 helix stability by translational purine riboswitches is the ligand-responsive reconfiguration of the residues in the 3WJ. For transcription-regulating purine riboswitches (e.g., *pbuE* riboswitch), it is unknown how the co-transcriptional binding of the ligand may influence the reaction-state kinetics or whether regulation is achieved via a different mechanism. However, the 3D architecture, including tertiary interactions and the 3WJ, is highly conserved among the transcriptional and translation purine riboswitches with known structures [15,16,17,18,19]. The primary difference in the transcriptional regulators is the higher GC content among P2 and P3 duplexes [16], the altered stability of which could play a mechanistic role in the kinetics of conformational switching.

### 2.2. Flavin Mononucleotide (FMN) Riboswitch

One of the most structurally unique riboswitches is the FMN riboswitch, whose aptamer comprises six helical domains interconnected by a six-way junction that forms the FMN-binding pocket [21,22,23]. Although this aptamer structure is more complex than that of the purine riboswitches, its near-symmetrical fold can be understood as the combination of two fused domains, each with its own 3WJ architecture, oriented perpendicularly (Figure 2A). One domain is formed by P1, P2, and P6 (domain I) and the other by P3, P4, and P5 (domain II). This generates a rather inflexible scaffold, whereby only the protruding helices, P1 and P4, have any substantial degree of mobility. In this respect, P4 is analogous to P1 and serves as a pseudo-regulatory helix in domain II. The existence of two regulatory helices forms the basis for ligand-induced conformational switching, which is achieved via mutually exclusive coaxial stacking within each domain [23]. The switchyard is centered around the FMN-binding site and involves a triad of adenine residues: A48, A49, and A85. In the ligand-bound conformation, the isoalloxazine ring of FMN is intercalated between the nucleobases of A48 and A85, while also forming T-shaped π–π interactions with A49. As such, FMN facilitates the continuity and coaxial alignment of P1/P6 (domain I) and results in the misalignment of P3/P4 (domain II). Without a ligand, the gap between A48 and A85 enables the more favorable coaxial stacking of P3/P4, as A48 rotates by ~45°. This, ultimately, results in the subtle misalignment of P1/P6 and the destabilization of P1.

### 2.3. Glutamine Riboswitch

In addition to riboswitches that recognize nucleobases and metabolic cofactors, others recognize amino acids and are involved in regulating their biosynthesis. One of these is the L-glutamine (Gln)-sensing riboswitch. Although the aptamer of this riboswitch exhibits the general three-helix Y-shaped architecture, it is unique in several respects. First, ligand binding stabilizes the aptamer in an open conformation, rather than closed (Figure 3A). In other Y-shaped aptamers, ligand binding induces or stabilizes the tertiary interactions between the P2 and P3 helical arms. In this case, however, the A-minor tertiary interactions between P2 and P3 are disrupted upon ligand binding, as the two helices separate from one another. Second, the 3WJ and ligand-binding site forms at the base of the P1 stem, rather than above it (Figure 3B). Gln binds directly in the major groove of P1, seated at the P1–P3 interface through a network of hydrogen bonds with Mg^2+^-coordinated water molecules [24]. Ligand specificity is achieved through hydrophobic stacking interactions with the ligand side chain, which is wedged between J2/3 residues G22 and G23, as well as two hydrogen bonds with the amide group. Importantly, the junction harboring G22 and G23 is disordered in the absence of a ligand and is stabilized only through ligand-mediated interactions with P1. These include the formation of two base triples and the insertion of G23 into the P1 helix to form a linchpin Watson–Crick base pair with C60 (Figure 3B). In this way, the stability of the P1 helix is regulated through the ligand-dependent recruitment of junction residues to the base of the P1 stem and occurs at the expense of distal tertiary interactions.

### 2.4. Tetrahydrofolate (THF) Riboswitch

The THF riboswitch very much resembles the purine riboswitches in both size and shape [25,26,27,28]. There are a few key distinctions, however, which give rise to very different mechanisms in regulating P1 helix stability. The first major difference is that the THF riboswitch contains two cooperative binding sites. A short helix (P4) forms a 3WJ with P2 and P3 and creates a binding site that is quite far from P1 (Figure 4A). As the apo and holo structures of this riboswitch show very little difference in this region, it is likely that THF binds to this site co-transcriptionally and simply stabilizes the tertiary interaction between P2 and P3 [25,26,27,28]. A second binding site resides at the tightly packed 3WJ of P1, P2, and P3 and is formed via a pseudoknot between L3 and J1/2 (Figure 4A). The binding of THF to this site is more regulatory in nature, since the structure of the pseudoknot has a direct influence on the P1 stability. The pterin ring of THF stabilizes the pseudoknot through an extensive network of hydrogen bonds and π-stacking interactions and forms a base tetrad with U7, U35, and U42 (Figure 4B). These interactions are crucial in maintaining the coaxial alignment of P1/L3 with P3. Without a ligand, the base triple comprising A8, A34, and G44 at the P2/P3 interface is disrupted (Figure 4B). A34 and G44 pull away from the 3WJ as the PK unwinds via the rotation and translation of P1/L3 with respect to P3, resulting in a severe kink in the major groove between L3 and P3 (Figure 4A) [28]. It is this loss of coaxial stacking that ultimately results in P1 instability.

### 2.5. Thiamine Pyrophosphate (TPP) Riboswitch

The TPP riboswitch, also known as the *thi*-box riboswitch, is among the first riboswitches ever reported [1]. It is widespread and found in all three life domains, including bacteria, archaea, fungi, and plants. Most notably, it is the only riboswitch identified so far in eukaryotes [29,30]. In addition to transcription and translation, one of the unique features of eukaryotic TPP riboswitches is that some regulate gene expression through alternative splicing [29,30]. The crystal structures of the ligand-bound TPP riboswitches from bacteria and plants are remarkably similar [31,32], despite using regulatory mechanisms of transcription termination and alternative splicing, respectively. Their conformational differences, therefore, may manifest only in the unliganded state. The crystal structure of the *E. coli* TPP riboswitch in such an apo conformation was determined recently, providing the structural basis for the regulation of P1 helix stability [33]. Like most classes of P1 helix-regulating riboswitches, the ligand-bound architecture of this riboswitch consists of three helical domains intersecting at a 3WJ, stabilized by a distal tertiary interaction (Figure 5A) [31,32]. Similar to the Gln riboswitch, however, this tertiary interaction is transient and ligand-controlled, with significant effects on the global structure of the aptamer, not just P1 [33].

Rather than at the 3WJ, TPP binds near the tertiary interface, quite distant from P1, and serves as a tether between the two sensory domains comprising P2/P3 and P4/P5, respectively (Figure 5B). Notably, the TPP riboswitch currently is the only P1 helix-regulating riboswitch whose ligand makes no direct contact with the 3WJ or P1. The stability of the tertiary structure is dependent on P4/P5 interactions with the two TPP-coordinating Mg^2+^ ions and the TPP 4-amino-5-hydroxymethyl-2-methylpyrimidine (HMP) ring intercalation between G42 and A43 of J3/2 (Figure 5B). Without a ligand, the loss of these interactions results in the dissociation and unraveling of P5, yielding an ensemble of open and closed aptamer conformations. One such open conformation was captured in the crystallographic state, revealing a Y-shaped structure with completely splayed sensory domains. Similar Y-shaped apo conformations of a TPP riboswitch fused to a group II intron were observed in a recent cryo-EM study [34]. After extensive 2D and 3D classification, a low-resolution structure could be derived from a small subset of data. The highly heterogeneous particle images of the TPP riboswitch in the absence of a ligand are evidence of the conformationally dynamic solution ensemble. The conformational stability of the tertiary structure afforded by the TPP ligand is propagated to P1 via the 3WJ, which comprises two stacked A-minor tetrads that facilitate coaxial stacking between P1 and P2 (Figure 5B). In the absence of a ligand, the extensive mobility of the sensory domains and the large structural changes in P4/P5 result in the disorder of J2/4 and disrupt the P1-stabilizing effect of the base tetrads.

**Figure 5 ijms-25-10682-f005:**
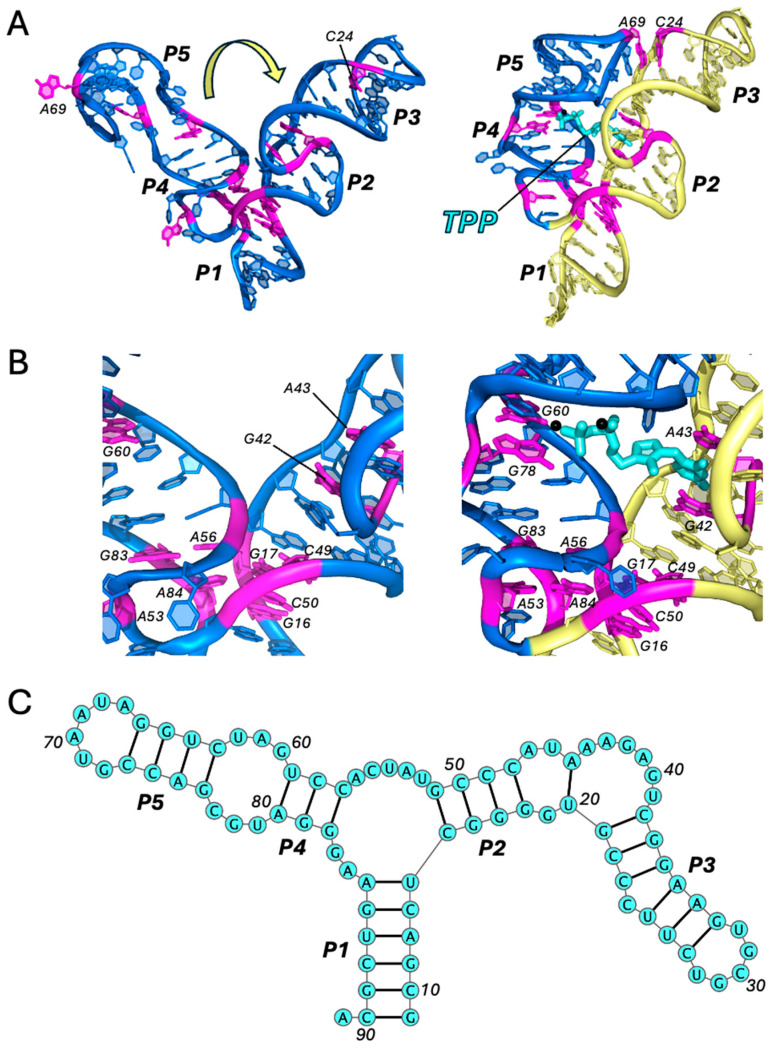
TPP riboswitch. Comparison of the apo (PDB ID: 8F4O) [33] and holo (PDB ID: 2GDI) [31] structures (all-atom RMSD 11.2 Å) of the TPP riboswitch aptamer (**A**) and its ligand-binding site (**B**). Ligand is shown in cyan. Key residues are colored in magenta. Regions of coaxial stacking that stabilize P1 upon ligand binding are colored in yellow. Motion of conformational change from apo to holo states is indicated by the yellow arrow. See also Movie 5. (**C**) Secondary structure (PDB ID: 2HOJ) [35], generated using VARNA (v3-93).

## 3. Apo Structures Resembling Ligand-Bound Conformations

The plethora of studies using various techniques, including X-ray crystallography, in-line probing, single-molecule Förster resonance energy transfer (sm-FRET), small-angle X-ray scattering (SAXS), NMR, cryo-EM, and MD simulations, has contributed immensely toward our understanding of the switching mechanisms of P1 helix-regulating riboswitches. The ligand-bound structures of these riboswitches all reveal P1 helices that are stabilized through ligand-facilitated coaxial stacking interactions that promote helical continuity with an adjacent helix. For many of these riboswitches, however, apo structures that provide a mechanistic basis for how these structural changes are regulated have yet to be elucidated. These include the lysine riboswitch, glycine riboswitch, cyclic di-GMP riboswitches, *S*-adenosylhomocysteine (SAH) riboswitch, and *S*-adenosylmethionine (SAM) riboswitches (SAM-I, SAM-III, SAM-IV, SAM-I/IV, SAM-VI), the ligand-bound structures of which have been determined and structurally characterized [36,37,38,39,40,41,42,43,44,45]. Of note, the crystal structures of the lysine and glycine riboswitches in the absence of a ligand were determined many years ago [36,37]. However, due to the buffer conditions and the effects of crystallization, these structures highly resemble their respective ligand-bound conformations. The same is true of the apo structure of SAM-IV (3.7-Å resolution), which was determined recently by cryo-EM [41]. As the buffer conditions in this study, particularly the Mg^2+^ concentration, were not reported, it is unknown whether the bound-like conformations were induced. It was also not clear how much the authors explored other conformations that may have been present in smaller populations. Lower-resolution cryo-EM volumes of the glycine and SAM-IV riboswitches in the absence of a ligand have also been reported [46], but mechanistic information pertaining to ligand-induced conformational changes could not be derived due to the similarity between the apo and bound structures and the lack of atomic details. Although the apo structures in bound-like conformations describe important ligand-induced changes locally, and very likely exist in the solution ensemble, they do not reveal the conformational changes associated with ligand binding that regulate P1 helix stability.

## 4. Future Implications

Given the difficulty of crystallizing flexible aptamers in the absence of a ligand without forcing them into bound-like conformations, cryo-EM should become the preferred method in exploring the conformational landscapes of riboswitches under biologically relevant conditions and in better elucidating their regulatory mechanisms. Importantly, cryo-EM provides greater potential for the structure determination of full-length riboswitches containing both the aptamer and expression domains. Such structural information is becoming increasingly obtainable given the rapidly growing number of cryo-EM structures of smaller and smaller RNAs. Cryo-EM structures with a resolution better than 10 Å have been reported for more than a dozen classes of RNA, many of which are smaller than 100 kDa [47]. Higher-resolution structures have been achieved using strategies that increase the sizes of single particles, such as RNA oligomerization-enabled cryo-EM via installing kissing loops (ROCK) [48] and the use of larger RNA scaffolds (reviewed in [49]).

At the most basic level, riboswitches must meet two criteria to achieve function: (1) the ability to recognize a specific ligand and (2) the associated conformational changes that control the fate of gene expression. It is quite obvious that the structures of aptamers will look different based on the ligand that each aptamer recognizes. However, the works summarized here on P1-helix-regulating riboswitches demonstrate further that these riboswitches also utilize different structural mechanisms for regulation. Thus, the fine-tuning of these riboswitches to respond to different metabolites with distinct chemical properties involves not only the structure that accommodates the ligand but also the structural changes that occur before (sensing) and upon (switching) the binding of this ligand. This fact highlights the significance of obtaining structural knowledge of aptamers in both apo and holo states. Information involving the structural versatility of RNA aptamers, including their mechanisms of action, is important not only in understanding the biology of naturally occurring riboswitches but also for the design of aptamer-based synthetic devices [10,50] and molecular biosensors [7,8,51]. Moreover, riboswitches continue to serve as viable targets for the development of novel antibiotics [52,53,54], which is of vital importance with the continual rise in antimicrobial resistance. The structures of riboswitches in their apo states may provide better targets for drug development than their ligand-bound counterparts. This approach could be especially effective for riboswitches that undergo large conformational changes to accommodate ligands. In such cases, a drug that stabilizes the aptamer in an inactive conformation could prevent the productive binding of the cognate ligand.

## Figures and Tables

**Figure 2 ijms-25-10682-f002:**
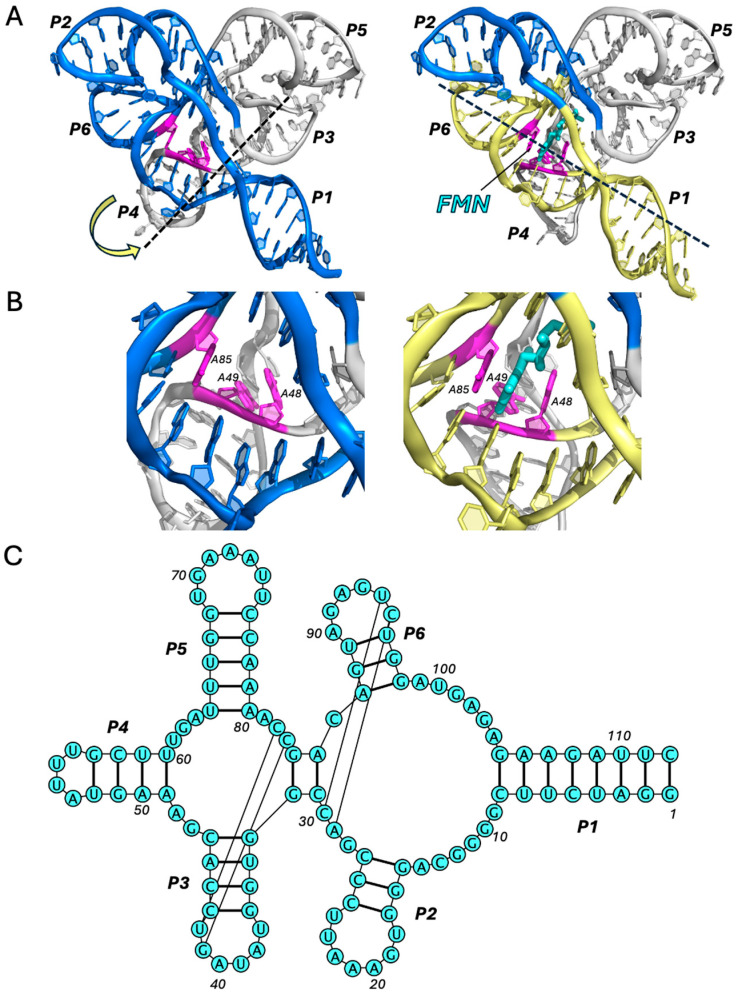
FMN riboswitch. Comparison of the apo (PDB ID: 6WJR) [23] and holo (PDB ID: 3F2Q) [21] structures (all-atom RMSD: 2.1 Å) of the FMN riboswitch aptamer (**A**) and its ligand-binding site (**B**). Domains I and II are colored in blue and gray, respectively. Ligand is shown in cyan. Key residues are colored in magenta. Regions of coaxial stacking that stabilize P1 upon ligand binding are colored in yellow. Motion of conformational change from apo to holo states is indicated by the yellow arrow. Black dotted lines indicate the direction of coaxial stacking in each state. See also Movie 2. (**C**) Secondary structure (PDB ID: 3F2Q), generated using VARNA (v3-93).

**Figure 3 ijms-25-10682-f003:**
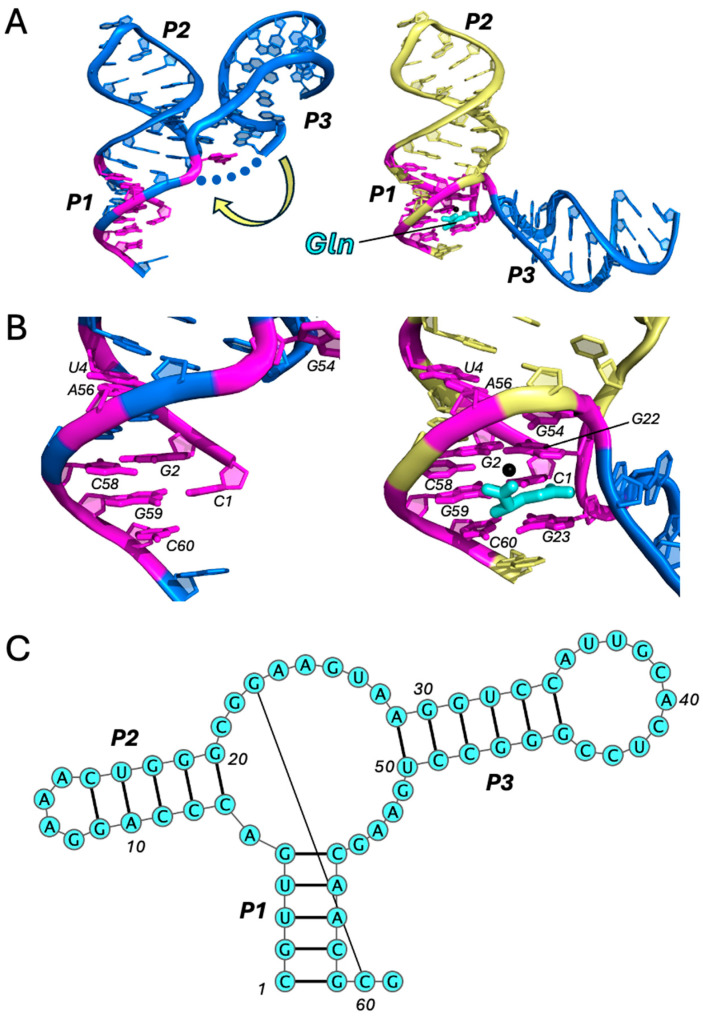
Gln riboswitch. Comparison of the apo (PDB ID: 5DDO) and holo (PDB ID: 5DDP) structures (all-atom RMSD 12.8 Å) of the Gln riboswitch aptamer [24] (**A**) and its ligand-binding site (**B**) Ligand is shown in cyan. Key residues are colored in magenta. Regions of coaxial stacking that stabilize P1 upon ligand binding are colored in yellow. Blue dots represent missing residues that are disordered in the crystal structure. Motion of conformational change from apo to holo states is indicated by the yellow arrow. See also Movie 3. (**C**) Secondary structure (PDB ID: 5DDP), generated using VARNA (v3-93).

**Figure 4 ijms-25-10682-f004:**
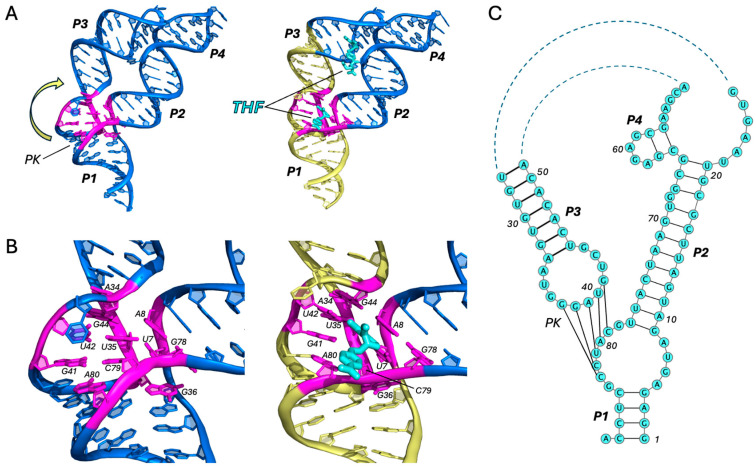
THF riboswitch. Comparison of the apo (PDB ID: 7KD1) [28] and holo (PDB ID: 4LVV) [25] structures (all-atom RMSD 3.6 Å) of the THF riboswitch aptamer (**A**) and its ligand-binding site (**B**). Ligand is shown in cyan. Key residues are colored in magenta, and the location of the pseudoknot (PK) is indicated. Regions of coaxial stacking that stabilize P1 upon ligand binding are colored in yellow. Motion of conformational change from apo to holo states is indicated by the yellow arrow. See also Movie 4. (**C**) Secondary structure (PDB ID: 4LVV), generated using VARNA (v3-93).

## Data Availability

All structural coordinates discussed in this review are available in the PDB databank under their respective accession numbers.

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
