# Peer review of "Riboswitch Mechanisms for Regulation of P1 Helix Stability"

_ijms, 2024, doi:10.3390/ijms251910682_

Round 1

Reviewer 1 Report

Comments and Suggestions for Authors

In this review, Stagno and Wang provide a summary of the structures of P1 helix-regulated riboswitches, including purine, FMN, glutamine, THF, and TPP riboswitches, in both ligand-free and ligand-bound states. Comparative analyses of these structures reveal common features, such as the 3WJ architecture, alongside distinct ligand-sensing characteristics and strategies for maintaining ligand-controlled P1 helix stability. This review enhances our understanding of riboswitch structure and stability regulation, potentially aiding in the discovery of drugs targeting these riboswitches. The review is well-organized, informative, and easy to comprehend. The figures and supplementary movies effectively illustrate the conformational changes of riboswitches upon ligand binding. Recently published papers on related topics are also referenced and discussed. The following concerns are raised for the authors to consider in revising and improving the manuscript.

Major:

After reading this manuscript, I felt there is a lack of introduction to the P1 helix and its regulatory role. A clear definition of P1 helix-regulated-riboswitches is also missing. This is a key concept for this review and is crucial for readers to fully appreciate the significance of ligand-controlled P1 helix stability.

Minors:

1.     Line 46-50: References need to be added.

2.     Line 145: The two 3WJ domains are oriented in perpendicular directions instead of opposite directions.

3.     Line 162: “_” before “of” should be deleted.

4.     Fig. 1B: Add zoom-in view of the ligand binding pocket from apo1 structure.

5.     Fig. 2C: Stems need to be labeled as in other figures.

Author Response

Reviewer 1

In this review, Stagno and Wang provide a summary of the structures of P1 helix-regulated riboswitches, including purine, FMN, glutamine, THF, and TPP riboswitches, in both ligand-free and ligand-bound states. Comparative analyses of these structures reveal common features, such as the 3WJ architecture, alongside distinct ligand-sensing characteristics and strategies for maintaining ligand-controlled P1 helix stability. This review enhances our understanding of riboswitch structure and stability regulation, potentially aiding in the discovery of drugs targeting these riboswitches. The review is well-organized, informative, and easy to comprehend. The figures and supplementary movies effectively illustrate the conformational changes of riboswitches upon ligand binding. Recently published papers on related topics are also referenced and discussed. The following concerns are raised for the authors to consider in revising and improving the manuscript.

Major:

After reading this manuscript, I felt there is a lack of introduction to the P1 helix and its regulatory role. A clear definition of P1 helix-regulated-riboswitches is also missing. This is a key concept for this review and is crucial for readers to fully appreciate the significance of ligand-controlled P1 helix stability.

We thank the reviewer for bringing up this important point. The term “P1 helix-regulated riboswitch” is now defined in the introduction (lines 65-71)

Minors:

  1. Line 46-50: References need to be added.

References had been added.

  1. Line 145: The two 3WJ domains are oriented in perpendicular directions instead of opposite directions.

The Reviewer is correct. The text has been changed to read “perpendicularly.”

  1. Line 162: “_” before “of” should be deleted.

The extra character has been deleted.

  1. 1B: Add zoom-in view of the ligand binding pocket from apo1 structure.

A new figure panel has been added.

 Fi 2C: Stems need to be labeled as in other figures.

Fig. 2C has been labeled accordingly.

Reviewer 2 Report

Comments and Suggestions for Authors

The manuscript titled "Riboswitch mechanisms for regulating P1 helix stability" provides a comprehensive examination of riboswitches, specifically emphasizing the structural mechanisms that govern the stability of the P1 helix when a ligand is bound. The authors present a thorough analysis of several riboswitch classes, skillfully contrasting their structural mechanisms among distinct types and demonstrating the intricacies of P1 helix control.

However, there are aspects in which the article may be enhanced. An area that stands out is the level of experimental comprehensiveness offered for each riboswitch example.  I recommend the authors consider including more up-to-date findings or developing methodologies in riboswitch research, namely cryo-EM. Furthermore, including up-to-date research on developing drugs that specifically target riboswitches, especially concerning antibiotic resistance, will improve the conversation and emphasize the therapeutic possibilities of these intriguing RNA structures.

Here are some insights for the authors to consider:

The recent cryo-electron microscopy (cryo-EM) developments have greatly improved our knowledge of RNA structures, particularly riboswitches. Although the article references cryo-EM, including more recent research that has employed cryo-EM to achieve higher resolutions in resolving riboswitch structures would be advantageous. An example is the 2024 study conducted by Lee et al., which provides valuable context to the discussion on structural determination methods by addressing the challenges and innovations in applying cryo-EM to small RNAs. The study can be found at https://pubmed.ncbi.nlm.nih.gov/38151772/ and https://www.nature.com/articles/s41392-022-00916-0.pdf. Access the following link: https://www.sciencedirect.com/science/article/pii/S0959440X24001210

The discipline of synthetic biology has made substantial advancements in using riboswitches as regulatory components. Reference to a 2024 publication in Nature Communications would offer readers a current perspective on the engineering of riboswitches for synthetic biology purposes, including gene expression regulation and biosensor development (https://www.nature.com/subjects/synthetic-biology/ncomms).

A prospective analysis can be drawn from the 2023 review in Trends in Microbiology provides an overview of current findings on novel riboswitch classes and their possible uses. https://pubmed.ncbi.nlm.nih.gov/38040620/ ; https://www.technologynetworks.com/drug-discovery/news/revolutionizing-drug-discovery-with-the-riboswitch-296738.

Comments on the Quality of English Language

Minor

Author Response

Reviewer 2

The manuscript titled "Riboswitch mechanisms for regulating P1 helix stability" provides a comprehensive examination of riboswitches, specifically emphasizing the structural mechanisms that govern the stability of the P1 helix when a ligand is bound. The authors present a thorough analysis of several riboswitch classes, skillfully contrasting their structural mechanisms among distinct types and demonstrating the intricacies of P1 helix control.

However, there are aspects in which the article may be enhanced. An area that stands out is the level of experimental comprehensiveness offered for each riboswitch example.  I recommend the authors consider including more up-to-date findings or developing methodologies in riboswitch research, namely cryo-EM. Furthermore, including up-to-date research on developing drugs that specifically target riboswitches, especially concerning antibiotic resistance, will improve the conversation and emphasize the therapeutic possibilities of these intriguing RNA structures.

Here are some insights for the authors to consider:

The recent cryo-electron microscopy (cryo-EM) developments have greatly improved our knowledge of RNA structures, particularly riboswitches. Although the article references cryo-EM, including more recent research that has employed cryo-EM to achieve higher resolutions in resolving riboswitch structures would be advantageous. An example is the 2024 study conducted by Lee et al., which provides valuable context to the discussion on structural determination methods by addressing the challenges and innovations in applying cryo-EM to small RNAs. The study can be found at https://pubmed.ncbi.nlm.nih.gov/38151772/ and https://www.nature.com/articles/s41392-022-00916-0.pdf. Access the following link: https://www.sciencedirect.com/science/article/pii/S0959440X24001210

We thank the reviewer for these thoughtful suggestions. The nature of this review focuses specifically on the structural mechanisms of conformational switching among P1 helix-regulating riboswitches. The riboswitches covered in the review are the only ones for which unliganded structures have been determined and their mechanisms elucidated, all using X-ray crystallography.  Therefore, expanding the comprehensiveness to include other structural methods would be to deviate from the structures presented, and thus the scope. We fully agree with the Reviewer, however, regarding the rise of cryo-EM as potentially the greatest tool to study riboswitch structure and mechanism in the future. This emphasis was made in the original version, but not extensively. We have thus reorganized the text to contain a new section titled, “Future implications,” the first paragraph of which highlights the developments and strategies toward obtaining high-resolution RNA-only structures by cryo-EM. These include methods such as ROCK and the incorporation of RNAs into larger RNA scaffolds to achieve larger particles and thus higher resolution structures.

The discipline of synthetic biology has made substantial advancements in using riboswitches as regulatory components. Reference to a 2024 publication in Nature Communications would offer readers a current perspective on the engineering of riboswitches for synthetic biology purposes, including gene expression regulation and biosensor development (https://www.nature.com/subjects/synthetic-biology/ncomms).

A prospective analysis can be drawn from the 2023 review in Trends in Microbiology provides an overview of current findings on novel riboswitch classes and their possible uses. https://pubmed.ncbi.nlm.nih.gov/38040620/ ; https://www.technologynetworks.com/drug-discovery/news/revolutionizing-drug-discovery-with-the-riboswitch-296738.

We agree, and we thank the Reviewer for these recommended citations. We have revised the last paragraph and added numerous references to studies/reviews on these important topics.